# Quantitative Relationship Analysis of Mechanical Properties with Mg Content and Heat Treatment Parameters in Al–7Si Alloys Using Artificial Neural Network

**DOI:** 10.3390/ma12050718

**Published:** 2019-03-01

**Authors:** Xiaoyan Wu, Huarui Zhang, Haiyang Cui, Zhen Ma, Wei Song, Weimin Yang, Lina Jia, Hu Zhang

**Affiliations:** 1School of Materials Science and Engineering, Beihang University, Beijing 100191, China; wuxiaoyan@buaa.edu.cn (X.W.); cuihaiyang@buaa.edu.cn (H.C.); mazhen@buaa.edu.cn (Z.M.); jialina@buaa.edu.cn (L.J.); zhanghu@buaa.edu.cn (H.Z.); 2Qingdao Research Institute of Beihang Universtiy, Qingdao 266000, China; songw@bhqditi.com; 3Sanmenxia Dicastal Co., LTD., Sanmenxia 472000, China; yangweimin0301@163.com

**Keywords:** artificial neural network, Mg content, heat treatment parameter, mechanical property, quantitative relationship

## Abstract

In this paper, an artificial neural network (ANN) model with high accuracy and good generalization ability was developed to predict and optimize the mechanical properties of Al–7Si alloys. The quantitative correlation formulas of the mechanical properties with Mg content and heat treatment parameters were established based on the transfer function and weight values. The relative importance of the input variables, Mg content and heat treatment parameters, on the mechanical properties of Al–7Si alloys were identified through sensitivity analysis. The results indicated that the mechanical properties of Al–7Si alloys were sensitive to Mg content and aging temperature. Then the individual and the combined influences of these input variables on the properties of Al–7Si alloys were simulated and the process parameters were optimized using the artificial neural network model. Finally, the proposed model was validated to be a robust tool in predicting the mechanical properties of the Al–7Si alloy by conducting experiments.

## 1. Introduction

Al–7Si alloys are extensively used in aerospace and automotive industries due to their excellent formability, high corrosion resistance, and good comprehensive mechanical properties [1,2]. The A356 and A357 alloys are two representative Al–7Si alloys with Mg contents ranging from 0.3 to 0.7 wt.% [2,3,4]. They are heat-treatable casting alloys, which can be strengthened by the precipitation of Mg_2_Si after T6 treatment [5,6]. The Mg element is a major alloying element for the formation of most effective hardening phase-metastable β″-Mg_2_Si phases. Higher Mg content increases the Mg–Si phase by altering the nucleation-driving force and growth kinetics. In addition, the size and morphology of the Mg–Si precipitates are controlled by the heat treatment process, including solution treatment, quenching, and artificial aging. Accordingly, an excellent strengthening effect benefits from the synergistic interaction between Mg content and heat treatment history.

Some investigations about the effect of alloying composition (especially Mg content) and processing parameters on the microstructure evolution and mechanical properties of Al–7Si alloys have been conducted. Yıldırım et al. [7] and Thirugnanam et al. [8] reported the effects of Mg amount on the intermetallics and properties of A356 alloys. Zhu et al. [9] and Long et al. [5] presented the effects of T6 heat treatment on a modified A356 alloy. However, the previous researches discussed the effect of Mg content and heat treatment parameters independently. The information of mutual interaction between Mg content and heat treatment parameters has not been shown. Moreover, the technique of optimizing the properties by trial and error leads to excessive wastage of manpower and money. 

With the emergence and development of the Materials Genome Initiative (MGI), the establishment of quantitative relationships between influence factors and properties provides an easy and efficient method in designing and optimizing alloys [10,11,12]: Prediction is prioritized and experimental verification comes later. Many mathematical models have been established to analyze the quantitative relationship [13,14,15]. However, it has been validated in Reference [16] that a mathematical model cannot predict the properties accurately due to the complex and nonlinear relationship between the variables and the properties.

In the past few decades, many artificial intelligence technologies [17,18,19] have been applied to describe this nonlinear relationship, such as genetic algorithm, neurocomputing, fuzzy logic, and artificial neural network (ANN). Compared with other methods, artificial neural network (ANN) [20,21] is a promising technology in the performance prediction field because it can solve problems faster. ANN models [16,22,23,24,25] are well known for function approximation and feature extraction of highly complex nonlinear relationships without any physical background knowledge. The technique has been widely applied in many aspects of materials science, including prediction of mechanical properties, modeling of the processing–microstructure–property relationships [22,26,27], modeling deformation behavior [28], and generation of processing maps for hot working processes [29]. All the above analyses indicated that ANN model can accurately predict and optimize the properties of aluminum alloy. However, quantitative relationship equations and the significance of each influence factor have not been proposed.

It is essential to attempt an ANN technique to enable the quantitative expression and understanding of the complicated nonlinear relationship of “Mg content–heat treatment parameters–properties” of the Al–7Si alloy. The aim of this work is to use an artificial neural network to empirically model and interpret the dependence of the mechanical properties of Al–7Si alloy on input variables of Mg content and heat treatment parameters. This in turn provides a method for designing the alloy with targeted mechanical properties. In this paper, an accurate ANN model was constructed and the quantitative relationship between the variables and mechanical properties of Al–7Si–xMg alloys were established. 

## 2. Experimental and Setup ANN Model

### 2.1. Experimental Process

In the present work, Al–7Si–xMg alloys were obtained using a low-pressure die casting technique at the same casting conditions. The compositions of the Al–7Si–Mg alloys with different Mg contents were monitored by direct reading spectrometry (Thermo ARL easySpark) and the corresponding chemical compositions are shown in Table 1. In order to investigate the effect of heat treatment parameters on Al–7Si alloys with different Mg contents, many samples with the size of 10 × 10 × 80 mm^3^ were cut from the castings at as-cast condition, and they were heat treated at a constant solution temperature of 535 ± 5 °C (which was close to the eutectic temperature) with different solution times and aging parameters. Then the heat treated samples were machined with the gage length of 30 mm and cross section diameter of 6 mm as shown in Figure 1. Tensile tests were conducted according to the ASTM B557 standard using an Instron 8801 universal electromechanical testing system equipped with Bluehill software and a ±50 KN load cell at ambient temperature (~20 °C). The gauge length of the extensometer was 25 mm and the ramp rate for extension was 1 mm/min. The value of the mechanical properties was the average of the three specimens. TEM observation were performed on a transmission electron microscope (TEM-2100F) at 200 kV to observe the morphology of the Mg–Si precipitates. The specific values of variables (Mg content and heat treatment parameters) and the mechanical properties (ultimate tensile strength (UTS), yield strength (YS), and elongation) are summarized in Table 2.

### 2.2. Artificial Neural Network Modeling

In the present study, a multilayer artificial neural network with a back-propagation (BP) learning algorithm was employed to simulate the relation between the tensile properties and the heat treatment parameters of Al–7Si–xMg alloys. This work was accomplished by using the neural network toolbox available with MATLAB software. There were three kinds of layers in the neural network: Input layer, hidden layer, and output layer. All the layers were made up with compute units and connected by transfer functions. In the present study, the inputs for the model were Mg content and heat treatment parameters. The outputs were the ultimate tensile strength (UTS), yield strength (YS), and elongation. The details of the neural network methodology and comprehensive treatments regarding ANN can be found elsewhere [24,25,30,31]. In the current study, a number of neural networks with different numbers of neurons in the hidden layer and different transfer functions were trained and tested to optimize the architecture. The transfer functions connecting these four layers were purelin, purelin, and tansig, respectively, as shown in Equations (1) and (2).
(1)purelin(n)=n
(2)$$\mathrm{tansig}\left(n\right)=\frac{2}{1+e^{-2n}}-1$$

The parameter n was determined by the weight matrix and threshold obtained by the ANN model. It was found that the optimum model with 4–10–11–3 architecture was adequate. Table 3 presents the parameters used in this ANN model. The artificial neural networks with the best correlations for ultimate tensile strength, yield strength, and elongation were established and the ANN model designed for this study is presented in Figure 2. Finally, the relative importance of input variables on the output parameters was calculated according to the method reported in literature [32].

## 3. Results and Discussion

### 3.1. ANN Performance Analysis

It has been verified that an optical artificial neural network should satisfy three important conditions [30]: Compendious architecture, high fitting accuracy, and good generalization ability. These three characteristics were evaluated individually in this work. 

The regression analysis was used to evaluate the fitting accuracy and the results are presented in Figure 3. As can be seen, the ANN model for mechanical properties can not only be well trained/validated (R = 0.92) but also accurately predict the test sets (R = 0.9). It showed an excellent correlation for the prediction of mechanical properties. This generalization ability, which means the ability to forecast unknown data, is the most important parameter of an ANN model. In this study, ten groups of alloy samples were taken to predict the strength and ductility, and the comparative results of predicted data and experimental data are shown in Figure 4. The smaller the standard deviation (proximity of input data to Y = T line), the more accurate the forecasting effect of the ANN model. The maximum absolute deviation was 4.75 MPa for UTS (Figure 4a) and 1.3% for elongation (Figure 4b). It indicated a convincing agreement between the predicted and experimental results for both strength and ductility. Consequently, the ANN model with high accuracy and generalization ability suggests that the predicted values could replace experimental values for deeper and further research. 

### 3.2. Sensitivity Analysis of Input Variables

In order to design the Al–7Si alloys with required mechanical properties through optimizing Mg content and heat treatment process, it was essential to understand the influential importance of these input variables on output parameters. The index of relative importance (I_RI_) of input variables on mechanical properties of Al–7Si–xMg alloy is presented in Figure 5. The larger the absolute value of I_RI_, the stronger the degree of relationship between the input and the output parameters. A negative value indicated that the output parameter was inversely related to the input variables [32]. The estimated I_RI_ values for Mg content and heat treatment parameters were positive for strength and negative in the case of elongation. Specifically, for the UTS and elongation, the order of relative importance of sensitive factors was aging temperature > Mg contents > aging time > solution time, and that for the YS was Mg contents > aging temperature > aging time > solution time. Therefore, the effects of Mg content and aging temperature on mechanical properties were more obvious than those of other parameters. This result was in accordance with the strengthening mechanism of Al–7Si–Mg alloys.

### 3.3. Prediction and Optimization of Mechanical Properties with Single Factors

The trained neural network model was applied to understand, evaluate, and predict the different correlations in aluminum alloys. In this study, the effect of the single factor of Mg content or aging temperature on mechanical properties was predicted in a reasonable range. 

Mg was an important alloying element in the Al–Si alloy which basically formed the strengthening precipitates–Mg_2_Si phase. Figure 6a,b shows the variation of mechanical properties with Mg content. The predicted strength increased with the addition of Mg, while the ductility decreased accordingly. When the Mg content increased from 0.2 to 0.4 wt.%, the strength improved from 320 to 340 MPa and the elongation decreased slightly from 10% to 9%. However, when the Mg content increased from 0.4 to 0.7 wt.%, the elongation decreased dramatically from 9% to 4%. This variation was highly consistent with previous studies reported by Chen et al. [4,33]. This was related to the increased precipitation strengthening of Mg–Si precipitates in Al–7Si alloy with high Mg content [34]. Nevertheless, these increased precipitates weakened the plastic deformation capacity, which in turn decreased the elongation. Therefore, superior comprehensive mechanical properties were obtained at a proper Mg content level of 0.3–0.5 wt.%. 

Figure 6c,d presents the simulated influence of aging temperature on mechanical properties. It was found that with the increase of temperature, the strength increased first and then decreased. The highest strength of about 350 MPa was obtained when the aging temperature was between 180 and 190 °C, and the elongation decreased from 13% to 5% when the aging temperature increased from 150 to 200 °C. Consequently, the best comprehensive mechanical properties were obtained at the condition of 160–175 °C. Increasing the aging temperature accelerated the aging process since the diffusion rate of Si and Mg atoms was improved; thus, the strengthening effect was enhanced [35]. 

### 3.4. Prediction and Optimization of Mechanical Properties with Several Factors

As stated in the introduction, the influence of Mg content and heat treatment parameters on mechanical properties is complex and interdependent. Mg content influenced the hardening capacity and heat treatment parameters played an important role in the precipitation kinetics [36,37,38]. It was of great importance to examine the synthetic influence of Mg content and heat treatment parameters on mechanical properties. Accordingly, mechanical properties were simulated as a function of the synthetic effects of Mg content and heat treatment parameters and the predicted results are presented in Figure 7 and Figure 8. The isograms of UTS and elongation were drawn based on these 3D prediction figures (Figure 7 and Figure 8) and the results are presented in Figure 9. 

It can be deduced from Figure 7 and Figure 8 that a proper value of Mg content and heat treatment should be used to obtain the desired comprehensive mechanical properties. According to the predicted results, the Al–7Si alloy with more than 0.4 wt.% Mg accompanied by aging temperature between 170 and 190 °C and aging time about 10 h was needed in order to obtain an alloy with a high strength of more than 340 MPa. As for the elongation, lower aging temperature and time was beneficial to obtain high elongation. In addition, at a higher aging temperature, the time to peak strengthening effect was shorted due to the improved diffusion rate of both Si and Mg atoms.

In order to satisfy the performance requirement of the workpieces under specific environmental conditions, the parameters can be chosen based on the isograms in Figure 9. The best comprehensive mechanical properties could be obtained under the scope suggested by the stars and arrows in Figure 9. In the present study, the method of designing materials by setting the specific properties first and then selecting the most suitable compositions and processing parameters was proposed and established. The method of improving ductility included decreasing Mg content, aging temperature, and aging time along with increasing solution time. In addition, the density of the isogram represented the gradient of the UTS and elongation. This isogram also confirmed the sensitivity of the mechanical properties to the variables. A wider variation range suggested less impact, which was in accordance with the I_RI_ value.

### 3.5. Quantitative Relationship between Mechanical Properties and Variables

The ANN model is a combination of a mathematical function and associated weights between inputs, hidden units, and outputs. The quantitative relationships between mechanical properties (UTS, YS, and Elongation) and input variables (Mg content, solution time, aging temperature, and aging time) were established, and formulae of the mechanical properties with the second hidden layer are shown in Equations (3)–(5).
(3)$$\mathrm{UTS}=\frac{2}{1+e^{-2\left(\sum_{i\;=\;1}^{11}UiHi\;+\;1.4638\right)}}-1$$
(4)$$\mathrm{YS}=\frac{2}{1+e^{-2\left(\sum_{i\;=\;1}^{11}YiHi\;-\;0.0986\right)}}-1$$
(5)$$E=\frac{2}{1\;+\;e^{-2\left(\sum_{i\;=\;1}^{11}LiHi\;-\;1.5439\right)}}-1,$$
where U, Y, and L are weight matrices shown in Table 4. H is the output of each neuron in the second hidden layer, which can be calculated by the quantitative relationship between the first hidden layer and the second hidden layer (Equation (6)).
(6)$$H_{i}=aE1+bE2+cE3+dE4+eE5+fE6+gE7+hE8+iE9+jE10+m_{i},$$
where E is the output of each neuron in the first hidden layer, which can be calculated by Equation (7). The letters (a–j) are the weight matrices and m is the threshold, and they are shown in Table 5.
(7)$$E_{i}=\alpha_{i}M1+\beta_{i}M2+\gamma_{i}M3+\delta_{i}M4+f_{i},$$
where M is the input parameter, f is the threshold, and the letters (α–δ) are weight matrices, and they are shown in Table 6.

### 3.6. Optimizing Processing Parameters

According to the above proposed ANN model, the mechanical properties of the Al–7Si alloy can be predicted and the Mg content and heat treatment parameters can be obtained for targeted mechanical properties. Accordingly, three kinds of properties containing high UTS (A), high elongation (C), and high comprehensive properties (B) were predicted and their corresponding composition and heat treatment parameters were obtained. The specific parameters are presented in Table 7. Then verification experiments were conducted correspondingly. Finally, the experimental and predicted properties were compared and the results are shown in Figure 10. The mean deviation (MD) as the evaluation criterion was calculated according to Equation (8).
(8)$$\mathrm{MD}=\frac{1}{N}\sum_{i}^{N}\left|d\left(i\right)-a\left(i\right)\right|,$$
where d(i) is the predicted data from the ANN model and a(i) is the experimental data from the actual tensile tests. *N* is the total number of data in the study. 

In this work, the MDs for UTS, YS, and elongation were 1.96, 0.88, and 0.53 and the corresponding relative percentages were 0.6%, 0.3%, and 5.9%, respectively. All the results not only verified the excellent generalization ability of the ANN model, but also indicated that the ANN model was efficient in predicting the mechanical properties of aluminum alloys.

## 4. Conclusions

In this work, a feed-forward neural network with back-propagation artificial neural network (BP ANN) model with high accuracy was established to predict the mechanical properties of Al–7Si alloy. 

(1)The parameters of Mg content and heat treatment process influenced the mechanical properties of the Al–7Si cast alloy. Based on the sensitivity analysis of the input variables, the sequence of the influences on the mechanical properties was established. The results showed that Mg content and aging temperature were two important parameters in determining the mechanical properties.(2)Through the optimized ANN model, the quantitative relationships between input variables (Mg content, solution time, aging temperature, and aging time) and mechanical properties (UTS, YS, and elongation) were established by three formulas.(3)Based on the predicted results of the ANN model, the Al–7Si alloy with more than 0.4 wt.% Mg accompanied by an aging temperature between 170 and 190 °C and an aging time of about 10 h was adequate in order to obtain an alloy with high strength more than 340 MPa.(4)Based on the optimized ANN model, a new way to design the Al–7Si alloy with targeted mechanical property was proposed.

## Figures and Tables

**Figure 1 materials-12-00718-f001:**
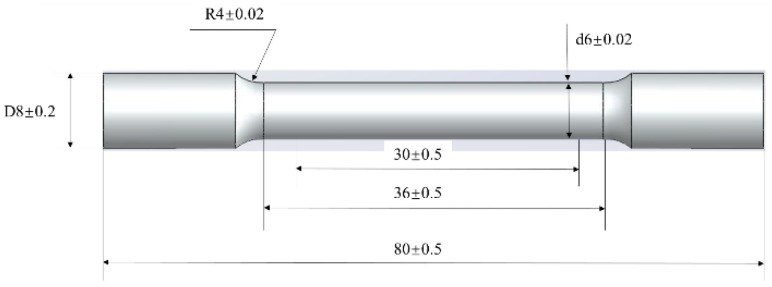
Schematic of tensile test specimens (mm) in the present study.

**Figure 2 materials-12-00718-f002:**
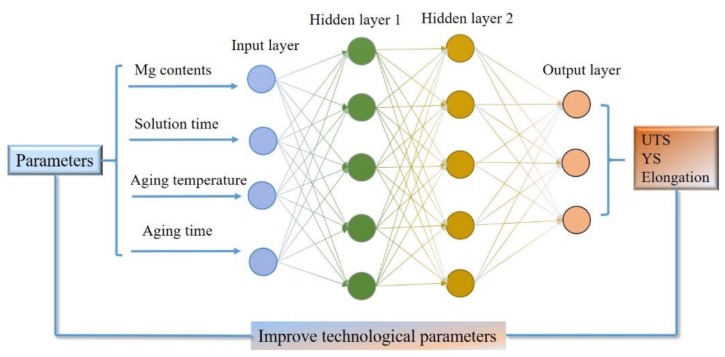
The artificial neural network designed for this study.

**Figure 3 materials-12-00718-f003:**
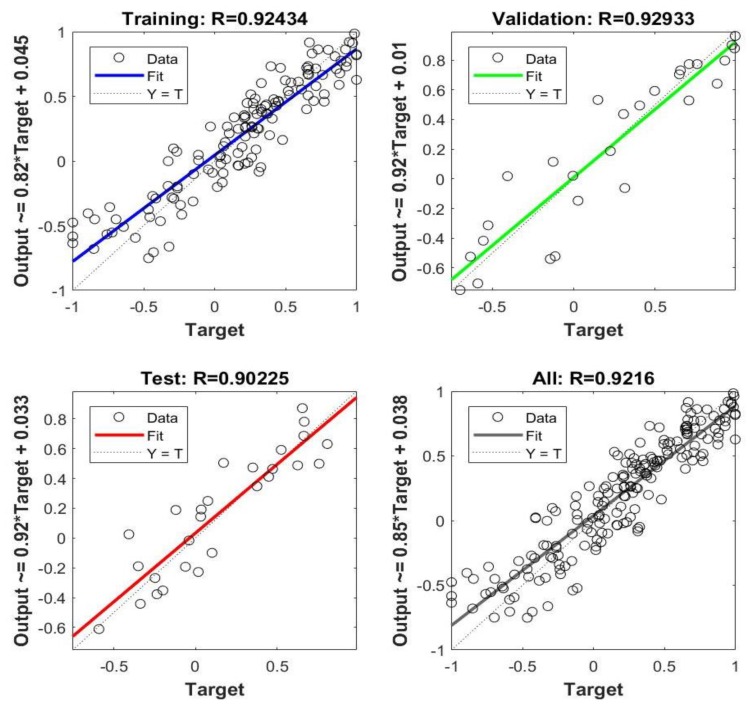
Regression analysis of the ANN model by program: R is the correlation coefficient.

**Figure 4 materials-12-00718-f004:**
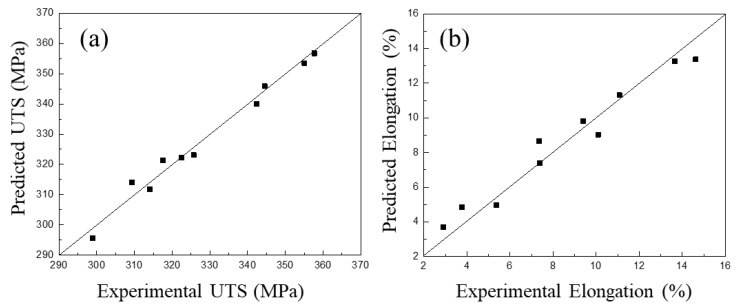
Comparison of the predictions and experiment results for: (**a**) Ultimate tensile strength (UTS) and (**b**) elongation.

**Figure 5 materials-12-00718-f005:**
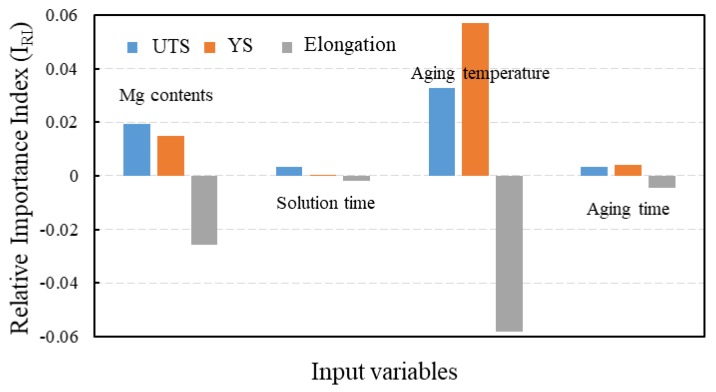
Bar charts showing the index of relative importance of Mg content and heat treatment parameters on UTS, yield strength (YS), and elongation of the Al–7Si alloy.

**Figure 6 materials-12-00718-f006:**
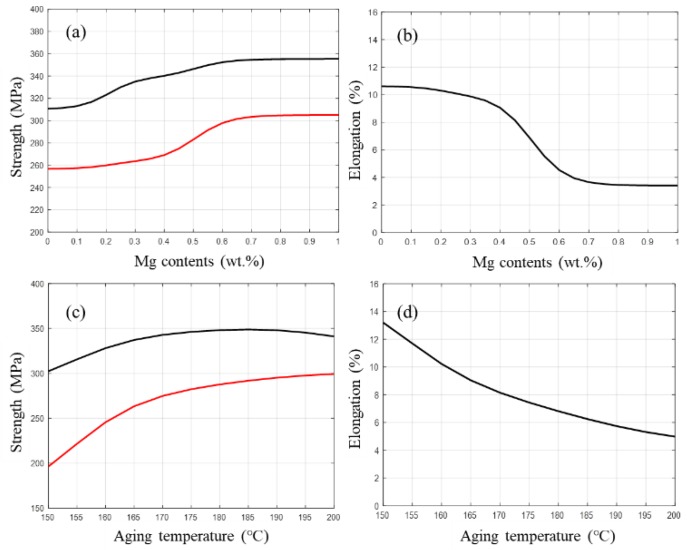
Predicted mechanical properties with (**a**,**b**) Mg content and (**c**,**d**) aging temperature.

**Figure 7 materials-12-00718-f007:**
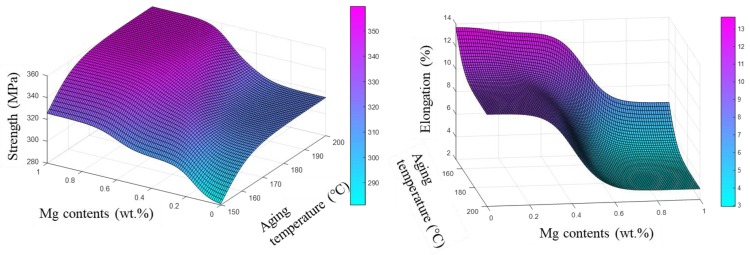
Predicted mechanical properties with Mg content and aging temperature.

**Figure 8 materials-12-00718-f008:**
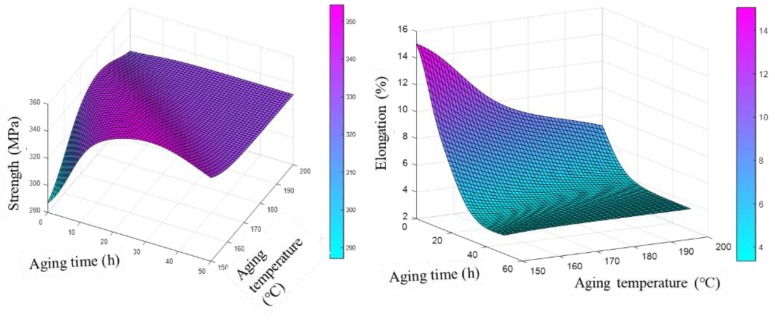
Predicted mechanical properties with aging parameters.

**Figure 9 materials-12-00718-f009:**
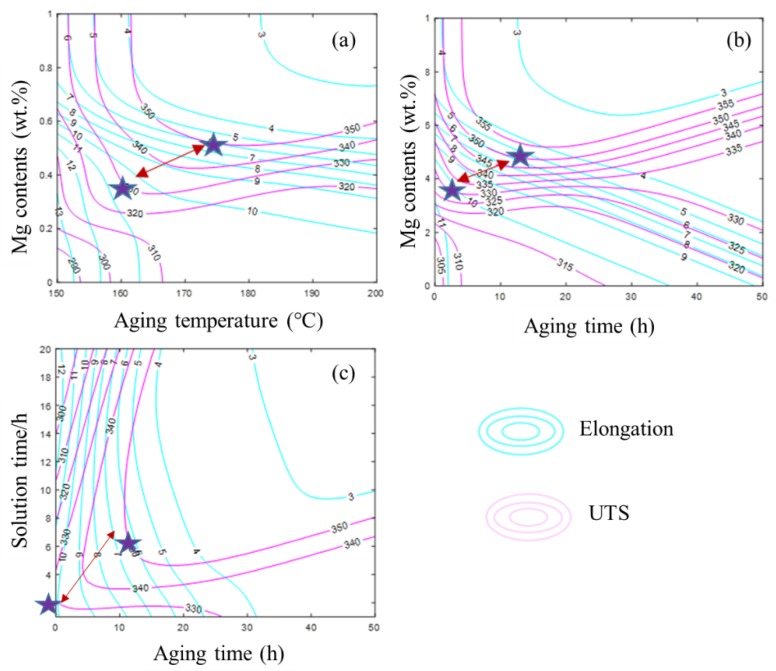
Isograms of UTS and elongation of Al–7Si alloy with different parameters: (**a**) Mg contents and Aging temperature; (**b**) Mg contents and Aging time and (**c**) Solution time and Aging time.

**Figure 10 materials-12-00718-f010:**
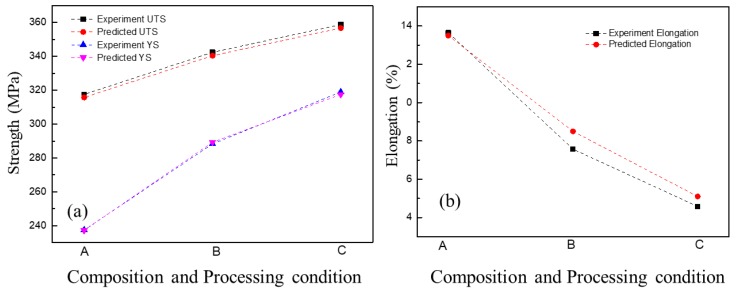
The comparison of experimental and predicted properties: (**a**) Strength and (**b**) elongation.

**Table 1 materials-12-00718-t001:** Chemical composition of the tested alloys (wt.%).

Alloys	Si	Mg	Fe	Cu	Ti	Sr	Al
Al–7Si–0.3Mg	7.134	0.301	0.115	0.072	0.144	0.015	Balance
Al–7Si–0.45Mg	7.123	0.455	0.122	0.079	0.148	0.015	Balance
Al–7Si–0.6Mg	6.978	0.608	0.111	0.075	0.15	0.015	Balance

**Table 2 materials-12-00718-t002:** Statistics of inputs and outputs used in the present model development.

Experimental Data	Input and Output Variables	Minimum	Maximum
65 training + 7 test data sets	Mg contents (wt.%)	0.3	0.6
	Solution time (h)	2	8
	Aging temperature (°C)	150	190
	Aging time (h)	1	42
	Ultimate tensile strength (MPa)	263.34	359.27
	Yield strength (MPa)	130.85	324.33
	Elongation (%)	1.07	18.19

**Table 3 materials-12-00718-t003:** The parameters of the proposed artificial neural network (ANN) model.

The number of layers	Input layers: 1, hidden layers: 2, output layers: 1
The number of neurons on the layers	Input neurons: 4, hidden neurons: 10 + 11, output neurons: 3
The initial weights and biases	Randomly between −1 and 1
The learning algorithm	Traindm
The learning rate	0.01
Activation function	purelin; purelin; tansig
Number of iterations	1000
Acceptable mean-squared error	0.001
The number of samples	72

**Table 4 materials-12-00718-t004:** The coefficient of the outer layer parameters and the second hidden parameters’ quantitative formulae.

	1	2	3	4	5	6	7	8	9	10	11
U	−0.4023	0.5292	−0.101	−0.0045	0.2112	0.5448	0.7152	−0.3718	−0.2403	0.6613	0.8859
Y	−0.6832	0.3263	−0.2466	0.5169	−0.5475	−0.3655	−0.3598	−0.6839	−0.2836	−0.2466	−0.4508
L	−0.2647	−0.0014	0.5874	0.2985	−0.2851	0.5568	0.3655	0.754	−0.4439	0.6605	−0.3456

**Table 5 materials-12-00718-t005:** The coefficient of the second hidden parameters and the first hidden parameters’ quantitative formulae.

	a	b	c	d	e	f	g	h	i	j	m
1	0.1355	0.1255	−0.9365	0.9695	0.6344	0.9022	0.6263	−0.2795	−0.1363	−0.1189	0.1248
2	0.0744	0.82	0.7178	0.8577	−0.6858	−0.0634	0.1839	0.4274	0.0599	0.9389	0.3639
3	0.2032	0.5231	−0.132	0.54	0.3878	0.291	0.7298	0.5249	−0.3397	−0.067	0.8802
4	0.7328	0.5411	−0.9638	−0.8818	0.6809	−0.3138	0.9492	−0.498	0.3938	−0.8847	0.2863
5	0.3375	0.5537	−0.4681	0.2553	0.784	0.0925	−0.8659	0.0174	0.914	−0.7707	0.0188
6	0.9318	−0.3483	0.1548	−0.1213	−0.8568	−0.0474	−0.1418	−0.4778	0.3957	−0.5603	0.5742
7	−0.2923	−0.8708	0.9148	0.0767	0.2453	0.4271	0.1244	−0.9487	0.6576	0.5976	0.9884
8	0.3035	0.8515	−0.9912	−0.3685	−0.0279	0.9527	0.5688	0.738	0.8249	−0.5653	0.9095
9	0.1694	−0.8473	0.6527	−0.0688	−0.4011	0.8883	0.0669	0.2996	0.728	0.8003	0.6363
10	−0.1005	−0.5128	0.9207	0.0134	−0.8876	−0.3215	0.4775	0.7924	−0.819	−0.1909	0.4992
11	−0.5446	−0.3827	0.4031	−0.6027	−0.3251	−0.5347	0.9466	0.8724	0.5415	0.0567	0.9977

**Table 6 materials-12-00718-t006:** The coefficient of the first hidden layer parameters and the input parameters’ quantitative formulae.

	α	β	γ	δ	f
1	0.1702	0.107	−0.7954	−0.8409	−0.5536
2	0.0846	0.8242	0.6174	0.0262	0.9687
3	−0.2316	0.8773	0.1259	−1.0369	−0.4795
4	−0.8426	0.2355	0.1741	0.5634	0.8002
5	−0.8532	0.8353	1.0346	0.3406	0.9818
6	0.4669	−0.3255	−0.5395	−0.6221	−0.4456
7	0.5046	0.8691	−0.9924	−0.3892	−0.0743
8	−0.6015	−0.0607	0.4271	−0.5255	−0.5005
9	−0.9256	−0.0907	0.2325	−0.5866	−0.2625
10	0.3276	−0.8509	−0.6511	−0.0639	−1.0178

**Table 7 materials-12-00718-t007:** ANN suggested optimum composition and heat treatment variables for desired mechanical properties.

	Designed Mechanical Properties			Mg Content/wt.%	Solution Time/h	Aging Temperature/°C	Aging Time/h
	UTS/MPa	YS/MPa	E/%				
A	315.8	237.2	13.5	0.27	3.85	164	2.2
B	340.4	289.3	8.5	0.45	4.2	168	17.5
C	356.8	317.5	5.1	0.65	3.3	173	12.8

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
