# Peer review of "Quantitative Relationship Analysis of Mechanical Properties with Mg Content and Heat Treatment Parameters in Al–7Si Alloys Using Artificial Neural Network"

_materials, 2019, doi:10.3390/ma12050718_

Round 1
Reviewer 1 Report
This work was nicely presented the close correlation between empirical data and simulated results for 2 variations of Al-7Si alloys.
However, I have a few comments and questions.
Please state what method was used to determine the chemical compositions presented in Table 1.
In Figure 4, you show 3 sets of data per graph. Please define what Y = T means in your text before you use it in your figure key.
In line 176 you state it is obvious that the density of precipitates increases with regards to Mg content. I would say that this is not a statement you can make with only 1 set of TEM micrographs. Please explicitly state (and ideally show) if the microstructural regions shown in Figure 7 are representative of the greater microstructure. Furthermore, in Figure 7, please label the zone axes used in Figures 7a and c. Additionally, the white arrows used to indicate precipitates in Figure 7 often point to areas with no visible precipitates. I recommend replacing these micrographs with new ones. Preferably replace these micrographs with ones taken in a two beam condition that makes the precipitates more distinct and also removes (or reduces) the contrast from the matrix due to it being aligned to a zone axis.
In Figure 10 please increase the font size of the all the text (besides the a, b, c labels).
Is there any work to test if this method works on other alloys besides the 2 variations of Al-7Si? If not that you should state that this method currently has only been used on these alloys. The final sentence is currently too broad. From the work presented it is difficult to state that this method indicated superior accuracy for Al alloys. Currently all that can be said is that this method works nicely for 2 alloys.
Please go through the entire manuscript to correct the numerous grammatical errors. It is recommended that a native English speaker is used to proof read the document.
Author Response
Dear Reviewer 1,
Thank you very much for your letter and your careful review and constructive suggestions on our manuscript “Quantitative relationship analysis of mechanical properties with Mg content and heat treatment parameters in Al-7Si alloys using artificial neural network”. We greatly appreciate your helpful comments, which are helpful for us to revise and improve our paper. We have studied the comments carefully and tried our best to revise the manuscript in detail. The grammar of revised manuscript also has been proofed by professional. Revised portion is marked in red in the manuscript. The point to point responses to your comments and the explanations regarding our revisions are listed in the text below. With these revisions, we hope our manuscript could meet standards of Materials and will be acceptable for publication in the Materials.
We appreciate for your warm work earnestly. Please feel free to contact us with any questions and we are looking forward to hearing from you soon.
Best regards
Huarui Zhang
School of Material Science and Engineering, Beihang University
37# Xueyuan Str., Haidian District, 100191, PR. China
Tel.: +86 10 82316482 Fax: +86 10 82338598
E-mail address: zhanghuarui@buaa.edu.cn

Reviewer 2 Report
I think the structure of the MLP (4-5-5-3) is over complicated. It was possible to train a neural network on a simpler combination. According to the heuristic rule of the geometric pyramid [Masters, T (1993). Practical neural network recipes in C++, Academic press, New York] for 4 inputs and 3 outputs, the rational design of neurons would be MLP (4-4-3-3) or MLP (4-3-3).
I would recommend inserting links to 2 articles:
Artificial intelligence monitoring of hardening methods and cutting conditions and their effects on surface roughness, performance, and finish turning costs of solid-state recycled aluminum alloy 6061 chips Abbas, A.T., Pimenov, D.Y., Erdakov, I.N., […], Soliman, M.S. DOI: 10.3390/met8060394 Metals: Volume 8, Issue 6, June 2018, 394
Minimization of turning time for high-strength steel with a given surface roughness using the Edgeworth–Pareto optimization method Abbas, A.T., Pimenov, D.Y., Erdakov, I.N.[…], Taha, M.A. DOI: 10.1007/s00170-017-0678-2 International Journal of Advanced Manufacturing TechnologyVolume 93, Issue 5-8, 1 November 2017, Pages 2375-2392
Author Response
Dear Reviewer,
Thank you very much for your letter and your careful review and constructive suggestions on our manuscript “Quantitative relationship analysis of mechanical properties with Mg content and heat treatment parameters in Al-7Si alloys using artificial neural network”. We greatly appreciate your helpful comments, which are helpful for us to revise and improve our paper. We have studied the comments carefully and tried our best to revise the manuscript in detail. The grammar of revised manuscript also has been proofed by professional. Revised portion is marked in red in the manuscript. The point to point responses to your comments and the explanations regarding our revisions are listed in the text below. With these revisions, we hope our manuscript could meet standards of Materials and will be acceptable for publication in the Materials.
We appreciate for your warm work earnestly. Please feel free to contact us with any questions and we are looking forward to hearing from you soon.
Best regards
Huarui Zhang
School of Material Science and Engineering, Beihang University
37# Xueyuan Str., Haidian District, 100191, PR. China
Tel.: +86 10 82316482 Fax: +86 10 82338598
E-mail address: zhanghuarui@buaa.edu.cn

Reviewer 3 Report
please see the attached file

Author Response
Dear Reviewer 3,
Thank you very much for your letter and your careful review and constructive suggestions on our manuscript “Quantitative relationship analysis of mechanical properties with Mg content and heat treatment parameters in Al-7Si alloys using artificial neural network”. We greatly appreciate your helpful comments, which are helpful for us to revise and improve our paper. We have studied the comments carefully and tried our best to revise the manuscript in detail. The grammar of revised manuscript also has been proofed by professional. Revised portion is marked in red in the manuscript. The point to point responses to your comments and the explanations regarding our revisions are listed in the text below. With these revisions, we hope our manuscript could meet standards of Materials and will be acceptable for publication in the Materials.
We appreciate for your warm work earnestly. Please feel free to contact us with any questions and we are looking forward to hearing from you soon.
Best regards
Huarui Zhang
School of Material Science and Engineering, Beihang University
37# Xueyuan Str., Haidian District, 100191, PR. China
Tel.: +86 10 82316482 Fax: +86 10 82338598
E-mail address: zhanghuarui@buaa.edu.cn

Reviewer 4 Report
The present paper can be of interest to community working with composite materials. However, it seems paper was written in hurry and it needs to be essentially improved before publication.
First of all there are too many grammatical errors: 1) Names and affiliations (highlighting the authors names is not common, and should be in front of last author and for affiliation - why all in capital ?)
2) Abstract: sentence "A correlation coefficient of over 90% was obtained." does not give any sense.
3) Introduction Ref. 8 => font should be changed - all should be written using same fond
4) once authors refers to equation they should first introduce those equations - not reverse way (they refer fist for equations given after)
5) they should use same equations style format for typing equations and also mentioning about, 6) sentence should not start with Fig. 6(c) and ... but as Figure 6(c) ...
7) Figures must be improved and re-plotted - some seems taken from elsewhere; axis descriptions are not readable -> e.g. Fig. 4(a) Predicted UTS/MPa it is incorrect should be Predicted UTS [MPa]. also Figs should be also labelled as Fig 2(a) etc figure captions should describe difference in each figs.
8) it would be important to provide partially the developed code in order other teams can use it.
Overall, present work must be essentially improved in order it can be published.
Author Response

(The authors gave the same response as above.)

Round 2
Reviewer 1 Report
Dear Authors,
Thank you for attempting to make the corrections I recommended.
It is unfortunate that you are not able to include higher resolution TEM micrographs because I think it would be a great benefit to your paper. However, I still suggest that you may wish to include some dark field micrographs to help make the precipitates a little more distinct in your matrix.
I have still found numerous grammatical errors and I recommend that you go through your entire paper once again and make corrections.
Author Response
Dear Reviewer 1,
Thank you very much for your letter and your careful review and constructive suggestions on our manuscript “Quantitative relationship analysis of mechanical properties with Mg content and heat treatment parameters in Al-7Si alloys using artificial neural network”. We greatly appreciate your helpful comments, which are helpful for us to revise and improve our paper. We have studied the comments carefully and tried our best to revise the manuscript in detail. The grammar of revised manuscript also has been proofed by professional.Revised portion is marked in red in the manuscript. The point to point responses to your comments and the explanations regarding our revisions are listed in the text below. With these revisions, we hope our manuscript could meet standards of Materials and will be acceptable for publication in the Materials.
We appreciate for your warm work earnestly. Please feel free to contact us with any questions and we are looking forward to hearing from you soon.
Best regards
Huarui Zhang
School of Material Science and Engineering, Beihang University
37# Xueyuan Str., Haidian District, 100191, PR. China
Tel.: +86 10 82316482 Fax: +86 10 82338598
E-mail address: zhanghuarui@buaa.edu.cn

Reviewer 4 Report
The paper has been revised. However, there is still a lot of work before I can recommend paper for publication, i.e. it does not match level of standard article.
As such major revision is still needed.
1) dont use first name in citing, i.e. Musa Yidrim et al. [7] should be Yıldırım et al. [7] .. replace in entire manuscript
2) Figure 1 - it is more for engineers not scientific figure => in captions should be given what units of length and other are used + diameter should be not fi (Greeks!!!) - there is standard description.
3) Equations 1 and 2 are given without any text - just no starting - it should be given for instance by ":' or type again Equation 1 is given by ... and, similarly, Eq. 2 reads ...
same for other equations in text
4) Fig. 3 captions should has brief explanation of what does R = ... means,
5) Figs. 4, 6, 8, 9, 10 and 11 - should be corrected if text/ MPa, ... means divided not unit!! use either comma or semi-comma or brackets.
6) Figure 6(a) and 6(b) should be Figures 6(a) and 6(b) - same for other text!!
Overall, it must be still essentially improved - please follow other published "materials" articles to enhance quality of paper.
Author Response
Dear Reviewer 4,
Thank you very much for your letter and your careful review and constructive suggestions on our manuscript “Quantitative relationship analysis of mechanical properties with Mg content and heat treatment parameters in Al-7Si alloys using artificial neural network”. We greatly appreciate your helpful comments, which are helpful for us to revise and improve our paper. We have studied the comments carefully and tried our best to revise the manuscript in detail. The grammar of revised manuscript also has been proofed by professional.Revised portion is marked in red in the manuscript. The point to point responses to your comments and the explanations regarding our revisions are listed in the text below. With these revisions, we hope our manuscript could meet standards of Materials and will be acceptable for publication in the Materials.
We appreciate for your warm work earnestly. Please feel free to contact us with any questions and we are looking forward to hearing from you soon.
Best regards
Huarui Zhang
School of Material Science and Engineering, Beihang University
37# Xueyuan Str., Haidian District, 100191, PR. China
Tel.: +86 10 82316482 Fax: +86 10 82338598
E-mail address: zhanghuarui@buaa.edu.cn
